# Autoregressive Adversarial Post-Training for Real-Time Interactive Video Generation

**Shanchuan Lin**[*]   **Ceyuan Yang**   **Hao He**[†]   **Jianwen Jiang**[‡]
**Yuxi Ren**   **Xin Xia**   **Yang Zhao**   **Xuefeng Xiao**   **Lu Jiang**
ByteDance Seed
https://seaweed-apt.com/2

## Abstract

Existing large-scale video generation models are computationally intensive, preventing adoption in real-time and interactive applications. In this work, we propose autoregressive adversarial post-training (AAPT) to transform a pre-trained latent video diffusion model into a real-time, interactive video generator. Our model autoregressively generates a latent frame at a time using a single neural function evaluation (1NFE). The model can stream the result to the user in real time and receive interactive responses as controls to generate the next latent frame. Unlike existing approaches, our method explores adversarial training as an effective paradigm for autoregressive generation. This not only allows us to design an architecture that is more efficient for one-step generation while fully utilizing the KV cache, but also enables training the model in a student-forcing manner that proves to be effective in reducing error accumulation during long video generation. Our experiments demonstrate that our 8B model achieves real-time, 24fps, streaming video generation at $736\times416$ resolution on a single H100, or $1280\times720$ on $8\times$H100 up to a minute long (1440 frames).

In recent years, the field of visual content creation has been transformed by the rise of foundation models for video generation [4, 78, 69, 44, 95]. These models have enabled a wide range of powerful applications, including text-to-video generation, image-to-video synthesis, and controllable video creation conditioned on various multi-modal signals.

Building on this progress, researchers are beginning to explore more ambitious applications. One exciting direction is using video generation models as interactive game engines and world simulators [93, 6, 67, 4]. Unlike offline video synthesis, interactive video generation requires the model to respond to user inputs in real time and continuously generate coherent video as the world evolves.

While diffusion models produce high-quality videos, they are very expensive for real-time interactive video generation. Early approaches applied diffusion models frame-by-frame [93, 111]. However, these approaches incur high redundancy due to the need to reprocess the context frames at every frame generation step. To address this, diffusion forcing [7, 108, 40, 23] introduced progressive noise to parallelize denoising across frames. Recent work further reduced inference costs by incorporating causal attention, KV caching, and step distillation [117, 75], with the current best model [117] achieving four denoising steps.

Meanwhile, token-based autoregressive generation—popularized by large language models (LLMs) [5, 1, 19]—offers an alternative. Models like VideoPoet [43] treat video generation as a next-token prediction task, which can straightforwardly leverage KV caching to improve generation

---

[*]Shanchuan Lin: Corresponding author: `peterlin@bytedance.com`

[†]Hao He: The Chinese University of Hong Kong. Internship at ByteDance Seed.

[‡]Jianwen Jiang: ByteDance Intelligent Creation Lab.

efficiency. However, per-token decoding remains sequential, limiting parallelism and making it difficult to meet real-time demands.

In this work, we aim to address the three core challenges of interactive video generation: (1) achieving real-time video generation throughput, (2) maintaining a low latency for interactive signals, and (3) enabling causal video generation of an extended duration. To this end, we explore adversarial training as a new paradigm and propose autoregressive adversarial post-training (AAPT) as an effective strategy for transforming a pretrained video diffusion transformer into a highly efficient autoregressive generator.

Our approach offers several advantages. First, it is fast. Our model autoregressively predicts each latent frame in a single forward pass (1NFE) while fully exploiting the KV cache. Our architecture design further enables 2× higher efficiency than equivalent diffusion-forcing models distilled to one step. Second, it maintains better quality over long durations. Our adversarial approach enables full student-forcing training, which mitigates error accumulation for long video generation. Furthermore, our student-forcing approach does not require paired ground-truth targets, allowing us to train long video generators and bypass the limitations of short-duration training data. This is important, as single continuous shots of tens of seconds are extremely rare in most datasets.

We demonstrate these benefits empirically. In terms of speed, our 8B-parameter model achieves real-time 24fps video generation at 736×416 resolution on a single H100 GPU, and 1280×720 resolution on 8×H100 GPUs, with a latency of only 0.16 seconds, substantially outperforming CausVid [117], a 5B model that operates at 640×352 9.4fps with a 1.30-second latency. In terms of duration, our model can generate continuous 60-second (1440-frame) video streams while fully utilizing the KV cache. This significantly exceeds the previous best one-step generator, APT [49], which supports only 49 frames.

Our experiments focus on the image-to-video (I2V) generation scenario, where the first frame is provided by the user, as most interactive applications adopt this setting. We showcase our method on two interactive applications—pose-conditioned virtual human generation and camera-controlled world exploration—where users can steer video generation in real time through interactive inputs. Evaluations show that our model achieves performance comparable to the state of the art.

# 1   Related Work

**One-Step Video Generation**   Early video generation models [3, 81] using generative adversarial networks (GANs) [18] can achieve fast generation using a single network evaluation. However, the quality, duration, and resolution are poor by modern standards. Diffusion models [28, 84] are the current state-of-the-art, yet their iterative generation process is slow and expensive. Generating a few seconds of high-resolution videos can take minutes. Existing research has attempted to reduce the inference cost by proposing more efficient formulations [51, 55, 35], samplers [59, 60, 90], architecture [109, 121, 120, 119, 98, 17, 66], caching [63, 53, 125], and distillation, *etc.* In particular, step distillation [74, 83, 82, 58, 49, 48, 72, 116, 115, 77, 76, 97, 50, 55, 110, 8, 61, 56, 112, 42, 37] emerges as one of the most effective approaches and has been widely studied in the image domain and is also adopted in video models. Seaweed [78] and FastHunyuan [15] report that the generation of 5-second 1280×720 24fps videos can be distilled to 8 or 6 steps without much degradation in quality. For further reduction in steps, SF-V [123] and OSV [64] explore 2 seconds of 1024×768 7fps image-to-video generation using only a single step. Recently, APT [49] achieves real-time text-to-video generation of 2-second 1280×720 24fps videos on 8×H100 GPUs using a single step. This has inspired more downstream applications to explore one-step video generation [99, 11]. Our method extends adversarial post-training (APT) to the autoregressive video generation scenario.

**Streaming Long-Video Generation**   Early research in streaming and long video generation [26, 41, 96] applies training-free or pipeline approaches on small-scale image and video generation models but is limited in quality. Modern large-scale video diffusion models, *e.g.* MovieGen [69], Hunyuan [44], Wan2.1 [95], and Seaweed [78], adopt transformer architecture and are trained on much higher resolutions and frame rates. However, due to the quadratic increase in attention computation, these models are commonly trained to only generate videos up to 5 seconds. To support long-video generation, these models are also trained on the video extension task, which gives the model the first few frames as a condition. At inference, this allows the model to extend the generation and stream the result to users as 5-second chunks. The extension can only be performed a few times before the

error accumulation catches up. Recent works have also explored architectures with linear complexity to directly generate long videos [98, 17, 66], but they are not designed for streaming applications.

More recently, diffusion forcing [7] has been proposed for video generation. It assigns progressive noise levels to frame chunks so the decoding proceeds in a causal streaming fashion. Earlier work uses bidirectional attention [108, 40]. Recent works have moved toward causal attention with KV cache [117, 9, 75, 23]. Most notably, SkyReel-2 [9] and MAGI-1 [75] are diffusion-forcing video generation models trained from scratch. CausVid [117] explores converting existing bidirectional video diffusion models to causal diffusion-forcing generators. Some of these methods also apply step distillation to improve speed. MAGI-1 [75] distills the model to 8 steps and outputs 24 frames as a chunk. It reports real-time 1280×720 24fps generation on 24×H100 GPUs. However, this amount of computation limits wide adoption. CausVid [117] distills the model to 4 steps and outputs 16 frames as a chunk. It can generate 640×320 videos at 9.4fps on a single H100 GPU. In comparison, our method is significantly faster. Our model uses only a single step and achieves 24fps streaming at 736×416 resolution on a single H100 GPU, or 1280×720 on 8×H100 GPUs. Moreover, ours generates a single latent frame (4 video frames) at a time to minimize latency.

It is important to note that these diffusion-forcing models are still only trained up to a fixed-duration window, *e.g.* 5 seconds. Early approaches without KV cache can run a sliding window, but this becomes an issue for KV cache because the receptive field grows indefinitely. Applying a sliding window and dropping out KV tokens can't help because the remaining tokens in the cache were computed in the past and still carry the receptive field. Naive extrapolation at inference leads to out-of-distribution behaviors. Therefore, methods like CausVid [117], SkyReel-V2 [9], and MAGI-1 [75] still need to apply the extension technique at inference by restarting and re-computing some overlapping context frames to generate long videos. Except that the diffusion forcing objective naturally supports input tokens with different noise levels, so the context frames can be given as clean latent frames at the beginning, with no additional training necessary. However, this is not ideal as it causes wait time on real-world streaming applications. In contrast, our method supports streaming generations of minute-long videos using KV cache without stopping and reprocessing.

**LLMs for Video Generation**    Large language models (LLMs) [5, 1, 19] have widely adopted the causal transformer architecture [94] for autoregressive generation. Most notably, attention is masked to prevent attending to future tokens, the inputs are past predictions, and the output targets are shifted by one for predicting the next tokens. Recent research has shown that images and videos can also be generated in such an autoregressive fashion [87, 102, 106, 10]. Although causal generation with KV cache is computationally efficient, generating token-by-token prevents parallelization and is slow for high-resolution generation. Some research has explored the decoding of multiple tokens at once during inference [103, 71, 114], but there is a tradeoff for quality, and it is challenging to decode an entire frame at once. Our architecture is inspired by LLMs, but ours generates a frame of tokens at a time, trained using an adversarial objective. This is optimized for fast generation.

**Interactive Video Generation**    Our paper showcases our model's real-time interactive generation ability on two applications: pose-controlled virtual human video generation and camera-controlled world exploration. We briefly introduce the related works in each subfield.

Recent research has explored the use of video generation models to create interactive environments for gameplay and world simulation [2, 6, 93, 22, 67, 16, 13]. Typically, the first frame is given, and the model continuously predicts the next frame given user control (image-to-world). The control can be the discrete states in an action space or general-purpose camera position embeddings [24, 25]. However, the high computation cost of the existing video generation approaches greatly limits the resolution and frame rates. For example, GameNGen [93] and MineWorld [22] only generate videos around 320×240 resolution at 6∼20fps with small models of a few hundred million parameters. Recent works, *e.g.* Genie-2 [67], Oasis [13], Matrix [16], *etc.*, have moved toward large-scale architectures and higher resolutions. Though many report their methods can operate in real-time, the specific hardware requirements are not specified.

Interactive video generation also holds significant potential in the domain of virtual humans. Typically, the first frame is given to establish the identity, then the pose [30, 62] or other multimodal [34, 46, 45, 89] conditions are given to drive the subject. Existing works employ diffusion models with the extension technique to generate long videos [85]. The inference speed remains a major bottleneck that limits their applicability to offline human video generation tasks.

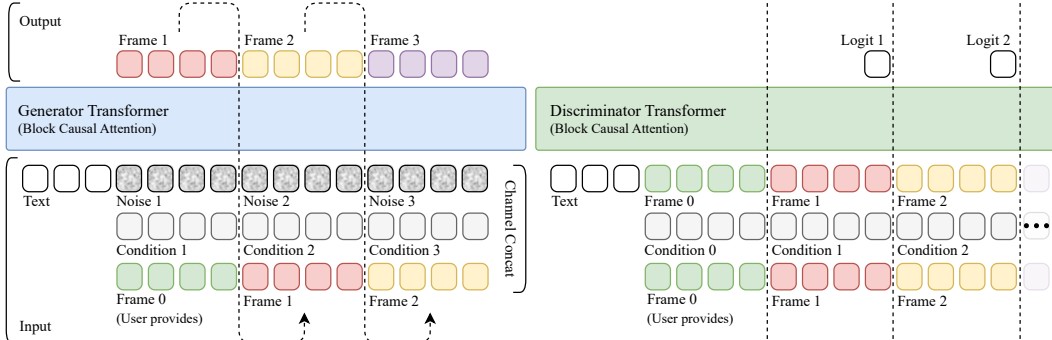

Figure 1: **Generator (left)** is a block causal transformer. The initial frame 0 is provided by the user at the first autoregressive step, along with text, condition, and noise as inputs to the model to generate the next frame in a single forward pass. Then, the generated frame is recycled as input, along with new conditions and noise, to recursively generate further frames. KV cache is used to avoid recomputation of past tokens. A sliding window is used to ensure constant speed and memory for the generation of arbitrary lengths. **Discriminator (right)** uses the same block causal architecture. Condition inputs are shifted to align with the frame inputs. Since it is initialized from the diffusion weights, we replace the noise channels with frame inputs following APT.

## 2 Method

Our objective is to transform a pre-trained video diffusion model into a fast, per-latent-frame causal generator suitable for real-time interactive applications. We achieve this through a new method called autoregressive adversarial post-training (AAPT). This section discusses AAPT's architectural transformations and training procedures.

### 2.1 Causal Architecture

We build our method on a pre-trained video diffusion model that employs a diffusion transformer (DiT) [68] architecture and operates in a spatially and temporally compressed latent space through a 3D variational autoencoder (VAE) [118]. Since our model operates in the latent space, we will refer to latent

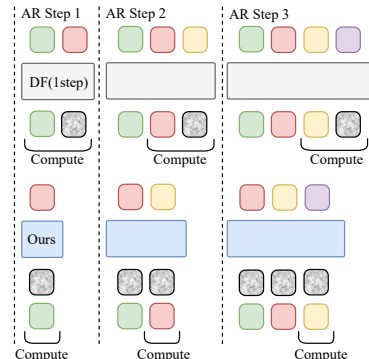

Figure 2: Ours is more efficient than one-step diffusion forcing (DF).

frames simply as frames unless otherwise specified. Our diffusion transformer has 8 billion (8B) parameters. It takes text embedding tokens, noisy visual tokens, and diffusion timesteps as input, and calculates bidirectional full attention over all the text and video tokens.

First of all, we transform the bidirectional DiT into a causal autoregressive architecture by replacing full attention with block causal attention. Specifically, text tokens only attend to themselves, and visual tokens attend to text tokens and visual tokens of previous and current frames. Afterward, we change the model inputs. As illustrated in Fig. 1, in addition to the regular noise and conditional inputs used by the original diffusion model, we change the model to also take in the past generated frame from the previous autoregressive step through channel concatenation, except the first autoregressive step where the input frame given by the user is used instead. During inference, our model runs autoregressively. At each autoregressive step, it reuses the attention KV cache and generates the next frame in a single forward pass. The generated frame is recycled, along with a new control condition, as inputs for the next autoregressive step.

To prevent the unbounded growth of attention computation and KV cache size, visual tokens attend to at most $N$ past frames while always attending to the text tokens and the first frame. It is worth noting that although each attention layer uses a window size of $N$, stacking multiple layers results in a much larger effective receptive field.

Our architecture resembles that of large language models (LLMs), but with one important distinction: unlike conventional next-token prediction that outputs the token probabilities using a softmax layer,

our model generates all tokens for the next frame in a single forward pass sampled by noise. In addition, our input recycling approach is also more efficient than the one-step diffusion forcing, as shown in Fig. 2. Diffusion forcing is not optimized for the one-step generation scenario. When using KV cache, diffusion forcing requires computation on two frames on every autoregressive step, while ours only needs one.

## 2.2 Training Procedure

To create a one-step, per-frame, autoregressive generator, our training process involves three sequential stages: (1) diffusion adaptation, (2) consistency distillation, and (3) adversarial training.

**Diffusion adaptation**    We load the pre-trained weights and finetune the model with the diffusion objective for architectural adaptation. We apply teacher-forcing training, where the ground-truth frames from the dataset are given as past-frame inputs. The output target is shifted by one frame to let the model perform next-frame prediction. Instead of pure noise, the noisy latent and the diffusion timestep $t \sim \mathcal{U}(0, T)$ are still used per regular diffusion training. The same noise level is applied for all frames. This resembles LLMs training, where all the autoregressive steps are trained in parallel.

**Consistency distillation**    We apply consistency distillation [83] before adversarial training as an initialization step to accelerate convergence following APT [49]. Our modified formulation and architecture are fully compatible with the original consistency distillation process without the need for modification. We omit classifier-free guidance (CFG) as we find it introduces artifacts in our autoregressive generation setting.

**Adversarial training**    We extend APT [49] to the autoregressive setting with improved discriminator design, training strategy, and loss objective.

For the discriminator model, we use the same causal generator architecture as our discriminator backbone, initialize it from the diffusion weights post-adaptation, and insert logit output projection layers. We replace the noise input to frames and randomly sample timestep $t \sim \mathcal{U}(0, T)$ for fast adaptation. A notable difference to APT discriminator design is that ours computes output logit for every frame instead of for the whole clip. This design naturally enables parallel multi-duration discrimination, as inspired by multi-resolution discrimination [39, 38].

We find models trained with teacher-forcing incur significant error accumulation at inference. To address this, we introduce a student-forcing approach within the adversarial training framework. Specifically, the generator only uses the ground-truth first frame and recycles the actual generated results as input for the next autoregressive step. In each training step, the generator is autoregressively invoked with KV cache to produce the video, exactly matching the inference behavior, while the discriminator evaluates all the generated frames in a single forward pass in parallel. We find detaching the pass-frame input from the gradient graph improves stability. We allow the gradient to flow through the KV cache to update all the parameters.

For the loss, we use R3GAN [31] objective as our preliminary experiments find that it is more stable than the non-saturating loss [18]. Specifically, we adopt the relativistic loss [36] and apply both the approximated R1 and R2 regularizations [73, 65] as proposed in APT [49].

**Long-Video Training**    For the model to learn continuous generation of long videos, one must train it on single-shot videos of long duration (*e.g.*, 30–60 seconds). However, such long single-shot videos are rare in most training datasets, where the average shot duration is only 8 seconds. The lack of long-duration training leads to poor temporal extrapolation during inference.

To address the data limitation, we let the generator produce a long video, *e.g.* 60 seconds, and break it down into short segments, *e.g.* 10 seconds, for discriminator evaluation. We keep an overlapping 1-second duration for discriminator evaluation to encourage segment continuation. The discriminator is trained on generated segments and real videos from the dataset. This objective ensures that every segment of a generated long video fits the data distribution.

To fit the GPU memory, we also let the generator only produce a segment at a time to be evaluated by the discriminator. To produce the next segment, the generator reuses the detached KV cache from the last segment. The gradient is backpropagated after every segment evaluation for loss accumulation.

This technique can be used to train very long generators, with the trade-off of an increase in training time. We find this technique significantly improves the quality of long-duration video generation. This is made possible by the discriminator in adversarial training. Unlike supervised objectives that require ground-truth targets, the discriminator does not need explicit supervision for each input frame. Instead, it learns to distinguish real videos from generated ones. As a result, the model can learn from every video sample, rather than relying on a limited number of long-duration videos.

### 2.3 Interactive Generation Applications

We first train a model for the general image-to-video generation task without interactive conditions. This allows us to evaluate the generation quality on standard benchmarks. We then train two separate models on the pose-conditioned human generation task and the camera-conditioned world exploration task. This allows us to evaluate the controllability using two distinct condition signals. For the pose-conditioned human video generation task, we extract and encode the human pose from the training videos and provide it as a per-frame condition to the model following [46]. Similarly, for the camera-conditioned world exploration generation task, we follow [25] to extract and encode the camera origin and orientation as Plücker embeddings, with a few modifications to have it better support causal generation. We use similar training datasets as used in these prior works [46, 25]. We refer readers to our supplementary materials for additional details on our architecture, implementation, and training parameters.

## 3 Evaluation

**Experimental Setups** We use causal 3D convolution VAE [118] to compress the video temporally by 4 and spatially by 8. Therefore, our model autoregressively generates 4 video frames. The first input frame is independently compressed as a latent frame by the VAE. Since our VAE is causal, it naturally supports streaming decoding. We use attention window size $N = 30$ to attend to 30 latent frames (5 seconds). Additional details on the training setup are provided in the supplementary materials.

**Baseline and Metrics** Following prior work [117], we evaluate our method on the standard VBench-I2V benchmark [32] on both 120-frame short-video generation and 1440-frame long-video generation. For comparison, we select CausVid [117], Wan2.1 [95], Hunyuan [44], MAGI-1 [75], SkyReel-V2, and our own diffusion model as baseline. These models are selected because CausVid is the state-of-the-art for fast streaming generation, and other models are available open-source video generation foundational models that support I2V. Note, CausVid is a closed-source model and only reports VBench-I2V for 120-frame 12fps generation. Wan2.1 and Hunyuan are bidirectional diffusion models that only support up to 120-frame generation. MAGI-1 and SkyReel-V2 are diffusion-forcing models that support arbitrary-length streaming decoding, so we include them for the 1440-frame comparison. Our model is evaluated and compared at 736×416 resolution. Additional inference settings and 1280×720 results are provided in the supplementary materials.

**Main Results** Figure 3 qualitatively compares our method on one-minute (1440-frame) video generation against SkyReel-V2, MAGI-1, and our diffusion baseline. All three of them exhibit strong error accumulation after 20 to 30 seconds. For our diffusion baseline, we experiment using a lower CFG scale or using rescale [47] but it does not mitigate the exposure problem and can further cause more structural deformation, so we keep it at CFG 10. We also show that our AAPT model trained on only a 10-second duration cannot generalize to long videos in Fig. 3d. Long video training is critical, as shown in Fig. 3e. Figure 4 shows more results of our model across subjects and scenes.

Table 1 shows that our method achieves competitive performance compared to the state-of-the-art methods on the quantitative metrics. For 120-frame I2V generation, AAPT improves frame quality score and image conditioning scores compared to the diffusion baseline and is the best across all compared methods. The frame quality improvement concurs with the findings in APT [49] that adversarial training can improve visual quality. AAPT has resulted in a slight decline in temporal quality score compared to the diffusion baseline, but is still above Wan and closely follows Hunyuan. We note that CausVid has an exceptionally high temporal quality score, likely because it was trained on 12fps data, which usually results in a higher dynamic degree than other 24fps models, and the dynamic degree score is the main differentiator for the overall temporal quality. For 1440-frame

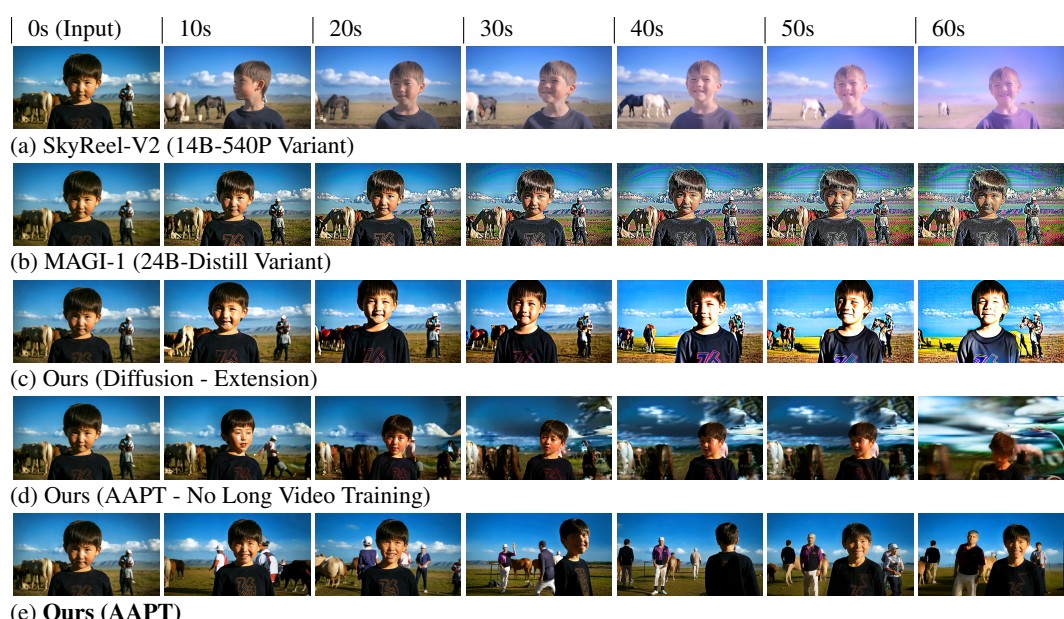

| 0s (Input) | 10s | 20s | 30s | 40s | 50s | 60s |

(a) SkyReel-V2 (14B-540P Variant)

(b) MAGI-1 (24B-Distill Variant)

(c) Ours (Diffusion - Extension)

(d) Ours (AAPT - No Long Video Training)

(e) **Ours (AAPT)**

Figure 3: Qualitative comparison on one-minute, 1440-frame, VBench-I2V generation.

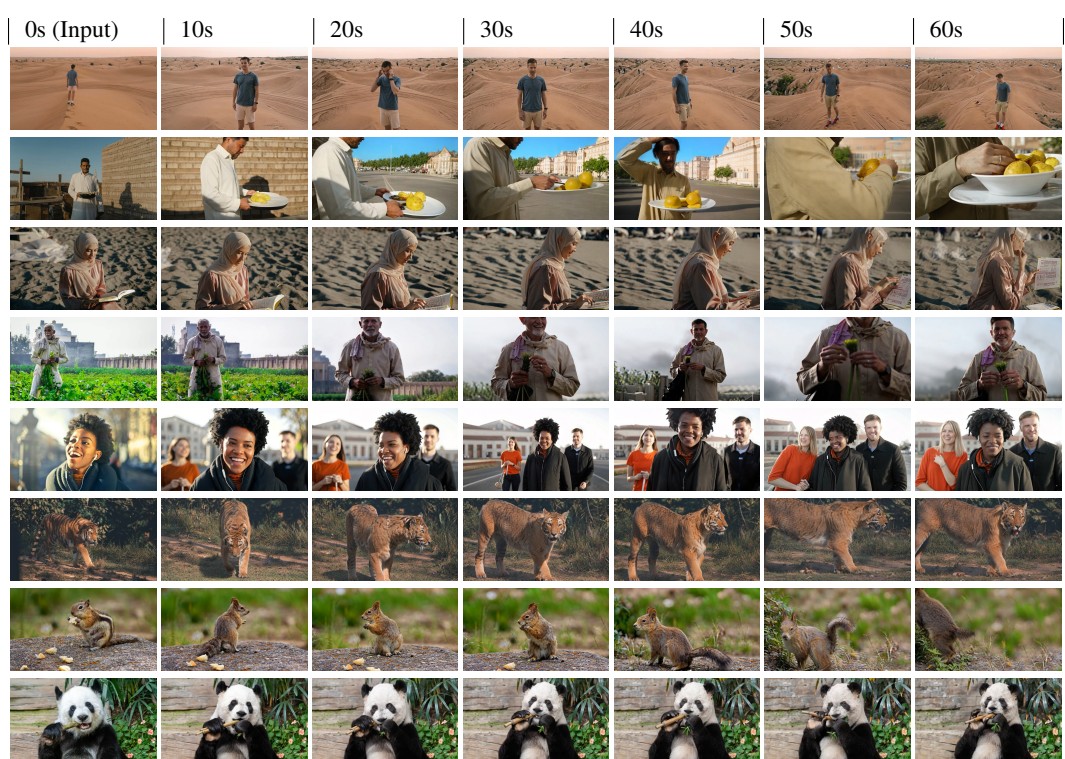

| 0s (Input) | 10s | 20s | 30s | 40s | 50s | 60s |

Figure 4: More results of our AAPT model for one-minute, 1440-frame, VBench-I2V generation.

I2V generation, AAPT achieves the best quality scores across the comparison and has improved conditioning scores compared to the diffusion baseline. We note that SkyReel-V2 and MAGI-1 have a higher image-conditioning score compared to our AAPT and diffusion baseline which is because most of the videos by MAGI-1 are stationary. This is reflected in its much lower dynamic degree score and the qualitative visualization in Fig. 3.

Table 1: Quantitative comparisons on VBench-I2V [32]. * denotes metrics that need special interpretation as discussed in the main text. The 6 quality metrics are aggregated as temporal quality and frame quality according to VBench-Competition. The best metrics are highlighted in bold.

| Frames | Method | Quality | | | | | | | | Condition | |
| | | Temporal Quality | Frame Quality | Subject Consistency | Background Consistency | Motion Smoothness | Dynamic Degree | Aesthetic Quality | Imaging Quality | I2V Subject | I2V Background |
|---|---|---|---|---|---|---|---|---|---|---|---|
| 120 | CausVid [117] | *92.00 | 65.00 | | | | Not Reported | | | | |
| | Wan 2.1 [95] | 87.95 | 66.58 | 93.85 | 96.59 | 97.82 | 39.11 | 63.56 | 69.59 | 96.82 | 98.57 |
| | Hunyuan [44] | 89.80 | 64.18 | 93.06 | 95.29 | 98.53 | 54.80 | 60.58 | 67.78 | 97.71 | 97.97 |
| | Ours (Diffusion) | 90.40 | 66.08 | 94.58 | 96.76 | 98.80 | 52.52 | 62.44 | 69.71 | 97.89 | 99.14 |
| | **Ours (AAPT)** | 89.51 | **66.58** | 96.22 | 96.66 | 99.19 | 42.44 | 62.09 | 71.06 | **98.60** | **99.36** |
| 1440 | SkyReel-V2 [9] | 82.19 | 53.67 | 78.43 | 86.38 | 99.28 | 47.15 | 53.68 | 53.65 | 96.50 | 98.07 |
| | MAGI-1 [75] | 80.79 | 60.01 | 82.23 | 89.27 | 98.54 | 25.45 | 52.26 | 67.75 | *96.90 | *98.13 |
| | Ours (Diffusion) | 86.65 | 60.49 | 82.38 | 89.48 | 98.29 | 66.26 | 56.46 | 64.51 | 95.01 | 97.72 |
| | **Ours (AAPT)** | **89.79** | **62.16** | 87.15 | 89.74 | 99.11 | 76.50 | 56.77 | 67.55 | 96.11 | 97.52 |

Table 2: Quantitative comparison on pose-conditioned human video generation task. Metrics better than ours are highlighted in bold.

| Method | AKD↓ | IQA↑ | ASE↑ | FID↓ | FVD↓ |
|---|---|---|---|---|---|
| DisCo | 9.313 | 3.707 | 2.396 | 57.12 | 64.52 |
| AnimateAnyone | 5.747 | 3.843 | 2.718 | 26.87 | 37.67 |
| MimicMotion | 8.536 | 3.977 | 2.842 | 23.43 | 22.97 |
| CyberHost | 3.123 | **4.087** | 2.967 | **20.04** | 7.72 |
| OmniHuman-1 | **2.136** | **4.111** | 2.986 | **19.50** | **7.32** |
| **Ours (AAPT)** | 2.740 | 4.077 | 2.973 | 22.43 | 11.78 |

Table 3: Quantitative comparison on camera-conditioned world exploration task. Metrics better than ours are highlighted in bold.

| Method | FVD↓ | Mov↑ | Trans↓ | Rot↓ | Geo↑ | Apr↑ |
|---|---|---|---|---|---|---|
| MotionCtrl | 221.23 | 102.21 | 0.3221 | 2.78 | 57.87 | 0.7431 |
| CameraCtrl | 199.53 | 133.37 | 0.2812 | 2.81 | 52.12 | 0.7784 |
| CameraCtrl2 | 73.11 | **698.51** | 0.1527 | **1.58** | **88.70** | 0.8893 |
| **Ours (AAPT)** | 61.33 | 521.23 | 0.1185 | 1.63 | 81.25 | 0.9012 |

Figure 5: Pose-conditioned virtual human

Figure 6: Camera-controlled world exploration

**Pose-Conditioned Human Video Generation** We evaluate our method on post-conditioned human video generation using the protocol and test set from previous work [45]. The pose accuracy is assessed via average keypoint distance (AKD) with keypoints extracted using DWPose [113]. For visual quality, we use Q-Align [107], a vision-language model, to evaluate image quality (IQA) and aesthetics (ASE). Additionally, Fréchet Inception Distance (FID) [27] and Frechet Video Distance (FVD) [91] measure the distributional alignment between the generated and the ground-truth samples. For comparison, we include four recent UNet-based diffusion models, *i.e.* Disco [101], AnimateAnyone [30], MimicMotion [122], CyberHost [45], and the state-of-the-art DiT-based OmniHuman-1 [46]. Table 2 presents the quantitative metrics. Among the six compared methods, ours is strong in pose accuracy and is ranked second only after the state-of-the-art baseline OmniHuman-1. In terms of visual quality, ours is consistently ranked second or third and is closely after CyberHost. Figure 5 shows visualization of our method.

**Camera-Conditioned World Exploration** We verify our method on the camera-conditioned world exploration task, also following the protocol of previous work [25]. We compute the FVD, movement strength (Mov), translational error (Trans), rotational error (Rot), geometric consistency (Geo), and appearance consistency (Apr). The details are provided in the supplementary materials. We compare against previous state-of-the-arts, *i.e.* MotionCtrl [104], can CameraCtrl 1 & 2 [24, 25]. Table 3 shows that our method achieves new state-of-the-art in three out of six metrics and closely follows CameraCtrl2 for the rest.

**Inference speed**    We compare the throughput and latency of our method to other streaming video generation methods in Tab. 4. Our method is significantly faster while achieving performance comparable to the state of the arts.

Table 4: Latency and throughput comparison.

| Method | Params | H100 | Resolution | NFE | Latency | FPS |
|---|---|---|---|---|---|---|
| CausVid | 5B | 1× | 640×352 | 4 | 1.30s | 9.4 |
| **Ours** | **8B** | **1×** | **736×416** | **1** | **0.16s** | **24.8** |
| MAGI-1 | 24B | 8× | 736 ×416 | 8 | 7.00s | 3.43 |
| SkyReelV2 | 14B | 8× | 960×544 | 60 | 4.50s | 0.89 |
| **Ours** | **8B** | **8×** | **1280×720** | **1** | **0.17s** | **24.2** |

## 4   Ablation Studies

**Long Video Training**    Table 5 reports VBench-I2V metrics on models trained with different durations for one-minute video generation. Specifically, the model trained for 60s significantly outperforms the model trained for only 10s, showing the effectiveness of long video training. Visualization is provided in Fig. 3d.

Table 5: One-minute generation performance using different training durations.

| Training Duration | Temporal Quality | Frame Quality |
|---|---|---|
| 10s | 85.86 | 57.92 |
| 20s | 85.60 | 65.69 |
| 60s | 89.79 | 62.16 |

**Teacher-Forcing and Student-Forcing**    Although diffusion adaptation and consistency distillation only support teacher forcing, adversarial training can be done in either teacher-forcing or student-forcing fashion. We describe the setup in the supplementary materials.

We find that models trained with teacher-forcing adversarial objective fail to generate proper videos at inference time, as shown in Figure 7. The content starts to drift significantly only a few frames into the generation process. Student-forcing training is critical in mitigating error accumulation. Although prior work has found that adding Gaussian noise to the input at training can reduce drifting at inference [93], it does not resolve the distribution gap from a fundamental level as student-forcing training. We leave additional explorations to future works.

**Limitations**    For consistency, our model can have difficulty maintaining the subject and the scene. This is caused by both the generator and the discriminator. Our generator adopts the basic sliding window for simplicity. We leave the exploration of more architectures and optimizations [105, 80, 52, 21, 20] to future works. For the discriminator, current segment-based discrimination cannot enforce long-range consistency. This may be mitigated by adding identity embeddings [54] to the discriminator. For train-

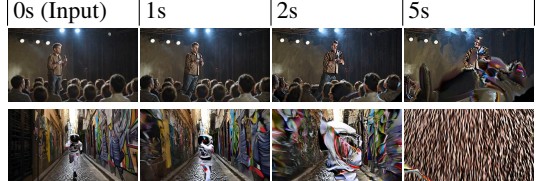

Figure 7: Models trained with teacher-forcing adversarial objective fail to generate proper content at inference.

ing speed, the long video training process can be slow. For quality, we find that one-step generation can still create defects, and once the defects emerge, they can be kept in the scene for a long time since the discriminator also enforces temporal consistency. More research is needed to improve the quality of one-step generation. For the duration, we test our model on zero-shot five-minute generation. Our model can still generate content but with artifacts. We provide examples in the supplementary material.

## 5   Conclusion

We have introduced autoregressive adversarial post-training (AAPT), a method that uses adversarial training as a paradigm to transform video diffusion models into a fast autoregressive generator suitable for real-time interactive applications. Our model achieves performance comparable to that of the best methods while being significantly more efficient. We also analyze its limitations and aim to address them in future work.

## Acknowledgment and Disclosure of Funding

We thank Weihao Ye for assistance with the evaluation. We thank Zuquan Song and Junru Zheng for assistance with the computing infrastructure. We thank Jianyi Wang and Zhijie Lin for their discussions during the work. This work is fully funded by ByteDance Seed.

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

# A    Model Architecture

**Diffusion Transformer**    Our diffusion transformer largely follows the MMDiT design [14]. It has 8B parameters and 36 transformer blocks. The discriminator adopts the same architecture. Therefore, our generator and our discriminator consist of 16B parameters for the adversarial training.

**Block Causal Attention**    We implement block causal attention using Flash Attention 3 [79] in a for-loop. We find it to provide reasonable performance for training. We leave the exploration for more performance implementation to future work. For inference, recurrent autoregressive steps are taken, and Flash Attention 3 can be naturally adopted without a performance penalty.

**Positional Embedding**    As the duration of the generation becomes agnostic to our causal architecture, we modify the 3D rotary positional embeddings (RoPE) [86]. Specifically, the positional embeddings continue to stretch dynamically along the spatial dimension to help the model generalize to different resolutions, while the positional embeddings are changed to have a fixed interval along the temporal dimension to support arbitrary lengths of training and generation.

**Parallelism**    We adopt FSDP [124] for data parallelism. We use ZERO 2 for the generator during student-forcing training that requires recurrent forward calls to avoid repeated parameter gathering, and ZERO 3 for all other modules to save memory. We also adopt Ulysses [33] as our context parallel strategy. We shard each video sample across 8 GPUs. Gradient checkpointing is also utilized per transformer block to fit the memory requirement.

# B    Training Details

**Diffusion Adaptation**    After changing the architecture to block causal attention and adding the recycled input channels, we first adapt the model with diffusion training.

We follow the original model to use the flow-matching parameterization [51]. Specifically, given sample $x_0$ and noise $\epsilon$, input is derived through linear interpolation $x_t = (1 - t) \cdot x_0 + t \cdot \epsilon$. The diffusion timestep is sampled uniformly $t \sim \mathcal{U}(0, 1)$, then passed through a shifting function $\text{shift}(t, s) := (s \times t)/(1 + (s - 1) \times t)$, where $s = 24$. Note that the same timestep is used for the entire clip without the diffusion-forcing [7] approach of assigning independent timesteps for each frame. Our model predicts the velocity $v = \epsilon - x_0$ and is penalized with the mean squared error loss. We apply the teacher-forcing paradigm and provide the ground-truth frames without noise as recycled input. The noisy input and the output target are shifted by one frame to facilitate next frame prediction.

We use AdamW optimizer [57] with a learning rate of 1e-5 and a weight decay scale of 0.01 throughout the process. We first train on 736×416 (equivalent to 640×480 by area) 5-second videos for 20k iterations with a batch size of 256. Then, we add 1280×720 to the mix for another 6k iterations with a batch size of 128. Finally, we turn up the maximum duration of 736×416 resolution videos to 15 seconds for 4k iterations with a batch size of 32. This curriculum allows our model to see enough samples in the early stages and see longer samples in the final stage.

**Consistency Distillation**    Then we apply consistency distillation [82] to create a one-step generator. Although the results after consistency distillation are blurry, it provides a better initialization for the adversarial training stage, as discovered by APT [49].

We inherit the same AdamW settings and the dataset settings as in the last diffusion adaptation stage. We distill the model on 32 fixed steps, which are uniformly selected and then passed through the shifting function with a shifting factor $s = 24$. We do not apply classifier-free guidance [29]. We continue to use the teacher-forcing paradigm to provide ground-truth frames as recycled inputs, and shift the noisy inputs and output targets by one frame following the diffusion adaptation approach. We follow the improved consistency distillation technique [82] and do not apply exponential moving average on the consistency target. No additional modification is needed for consistency distillation. The model is trained for 5k iterations.

**Adversarial Training**    Finally, we perform adversarial training. In this stage, we switch to the student-forcing paradigm, where the generator only takes the first frame as input and recycles the actual generated frame for the next autoregressive step, strictly following the inference procedure. Then, the discriminator evaluates the generated results in parallel, producing logits after each frame for multi-duration discrimination.

We follow APT [49] to initialize the generator from the consistency distillation weights, and to initialize the discriminator from the diffusion adaptation weight. We change to use the relativistic pairing loss [36]:

$$\mathcal{L}_{RpGAN}(x_0, \epsilon) = f(D(G(\epsilon, c), c) - D(x_0, c)), \tag{1}$$

where $G$,$D$ denote the generator and the discriminator respectively, $f_G(x) = -\log(1 + e^{-x})$ or $f_D(x) = -\log(1 + e^x)$ is used each of their update steps respectively, $c$ denotes the text condition and other interactive conditions. We calculate R1 and R2 regularization [73, 65] through the approximation technique proposed in APT [49]:

$$\mathcal{L}_{aR1} = \lambda \| D(x_0, c) - D(\mathcal{N}(x_0, \sigma\mathbf{I}), c) \|_2^2, \tag{2}$$

$$\mathcal{L}_{aR2} = \lambda \| D(G(\epsilon, c), c) - D(\mathcal{N}(G(\epsilon, c), \sigma\mathbf{I}), c) \|_2^2, \tag{3}$$

where $\epsilon = 0.1$ and $\lambda = 1000$. Since the discriminator is initialized from the diffusion model, we follow APT to provide timesteps by random uniform sampling $t \sim \mathcal{U}(0, 1)$. We do not shift the timestep for the discriminator. We use RMSProp optimizer with $\alpha = 0.9$ following APT [49].

We first perform training without the long-video extension training technique. The videos are 5s to 10s in duration. We train it using a low learning rate of 3e-6 following APT [49] and a batch size of 256 for 500 generator updates. The resulting model can only generate up to 10 seconds and will drift for videos longer than 10 seconds.

Then we apply the long video training technique. The training videos are still from 5s to 10s, and we extend it once with an overlap of 1s to a total maximum duration of 19s (10 + (10-1)). This stage is trained for 500 generator updates. Then we turn up the extension to 5 times, to a total maximum duration of 55s (10 + 5×(10-1)). We find it necessary to decrease the batch size to 64 and increase the learning rate to 1e-5 for the extension training for the model to make adequate changes in a reasonable amount of time.

Since the generator in student-forcing mode must recurrently perform model forward for each autoregressive step during training, we switch FSDP to ZERO 2 mode to save all the model parameters on each machine. This avoids repeated parameter gathering and improves the training seed. The discriminator and text encoder still adopt ZERO 3 to shard all the model parameters for memory saving.

**Computational Resources**  We use 256 H100 GPUs for our final training and employ gradient accumulation where necessary to reach our final batch size. The model is trained in approximately 7 days, where the diffusion adaptation and the long-video adversarial training take the majority of the time.

## C  Variational Autoencoder

We train a lightweight VAE decoder to fit the real-time budget. Specifically, our original VAE decoder has 3 residual blocks per resolution scale, and has channels [128, 256, 512, 512] at each resolution scale. Our lightweight VAE decoder reduces the number of residual blocks per resolution to 2, and reduces the channels to [64, 128, 256, 512]. This results in nearly 3 times speed-up without visible quality degradation.

## D  Teacher-Forcing Adversarial Training

The adversarial training supports both student-forcing and teacher-forcing modes. To implement student forcing, the generator runs autoregressively with KV cache and recycles the actual generated frame as input for the next autoregressive step. The discriminator evaluates the results in parallel. To implement teacher forcing, the generator takes ground-truth video frames as past prediction inputs and predicts the next frames in parallel. The discriminator runs autoregressively and always uses the KV cache from the real videos to attend to the ground-truth past frames.

Figure 8 visualizes teacher-forcing adversarial training. Specifically, in teacher-forcing mode, the generator given input $I1, I2, I3$ generates independent output $O2, O3, O4$. Namely, the output $O3$ only has a correlation with $I2$ but not with $O2$. Therefore, the discriminator must independently evaluate the generated results with their correct dependencies to produce logits $L2, L3, L4$. Since the discriminator transformer is causal, the repeated computation can be saved using KV cache.

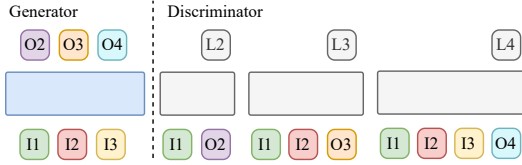

Figure 8: Teacher-forcing adversarial training

We have conducted experiments with teacher-forcing adversarial training, and the model fails to generate reasonable videos as discussed in the main paper. We suspect LLMs are able to train with teacher-forcing mode because they use a discrete codebook to encode words, where slight inaccuracy is less relevant. But our model predicts continuous latent values for the entire frame, where slight inaccuracy accumulates.

# E The Importance of Result Recycling

We conduct an experiment to study the importance of result recycling. Specifically, we keep the exact architecture and training settings, and we mask the recycled input as zero tensors, except the first frame, which takes in the user image. We find that models trained without recycling input cannot generate large motion. Some of the movements become incohesive as well. The video visualization is provided on our website.

# F I2V Evaluation

The table in the main text compares our model under the 736×416 setting. For the other models we compare to, we largely follow the default sampling setting for each model, including the number of steps and CFG [29]. We also use the default resolution for each model to ensure that the model has been properly trained on the expected resolution. Specifically, we use 896×544 for Hunyuan [44], 832×464 for Wan2.1 [95], 960×544 for SkyReel-V2 [9]. We note that we run 5 samples per prompt for all the comparisons per VBench-I2V [32] requirement, except for SkyReel-V2 which we only run 1 sample per prompt and reduce the sampling steps from its default 50 to 30. This is because SkyReel-V2 is too computationally intensive to generate one-minute videos.

We additionally provide the evaluation metrics under the 1280×720 resolution in Tab. 6. Note that 1280×720 is trained and inference with a smaller attention window size $N = 15$ to fit the memory.

Table 6: Quantitative VBench-I2V [32] metrics on 1280×720 compared to 736×416.

| Frames | Method | Resolution | Quality | | | | | | | | Condition | |
|---|---|---|---|---|---|---|---|---|---|---|---|---|
| | | | Temporal Quality | Frame Quality | Subject Consistency | Background Consistency | Motion Smoothness | Dynamic Degree | Aesthetic Quality | Imaging Quality | I2V Subject | I2V Background |
| 1440 | Ours | 736×416 | **89.79** | 62.16 | 87.15 | 89.74 | 99.11 | 76.50 | 56.77 | 67.55 | 96.11 | 97.52 |
| | | 1280×720 | 88.24 | **64.30** | 87.95 | 90.10 | 99.16 | 63.29 | 57.79 | 70.80 | **96.51** | **98.18** |

# G Camera-Conditioned World Exploration

**Training** We make a few modifications on CameraCtrl II [25] to make it better support causal generation. First, CameraCtrl II uses Plücker embeddings to represent the camera position and orientation, where it treats the first frame as the initial position, and the other frames are relative to the first frame. This is problematic as the value can grow unbounded if the displacement forever increases. We change it so that each frame is only relative to the previous frame. Hence, the Plücker embeddings only represent the camera changes between immediate frames to prevent unbounded growth of values. Second, CameraCtrl II uses the original Plücker coordinate to represent each camera ray, which consists of a direction vector and a moment vector. The moment vector encodes the displacement information, which is computed as the cross product of a point on the line and the direction vector. We find that this implicit representation unnecessarily increases the complexity for the model to learn. Rather, we directly encode the camera ray's origin and direction. Third, the input scaling to the model is in fact a hyperparameter that is not previously explored. We scale the coordinate inputs to roughly 1 standard deviation to simplify model learning. We also drop samples whose camera embeddings have very large values. These outliers are caused by inaccurate camera estimation and are detrimental to the stability of adversarial training. Last, we use random initialization instead of zero initialization for the input projection of the new channels. We find that random initialization helps the model to adapt to the new inputs much more quickly.

The camera-conditioned model is trained separately from the I2V model. We start from the I2V diffusion adaptation weights and continue training on the camera-conditioned task. The consistency distillation and adversarial training are done separately for this dedicated model. The training settings are mostly the same as the I2V model. For the long-video extension training, we randomly sample new camera trajectories for the extended parts.

**Evaluation** Our evaluation metrics follow CameraCtrl II [25]. Specifically, we compute Fréchet Video Distance (FVD) [92] against the ground-truth videos. We compute the movement strength (Mov) on RAFT-extracted [88] dense optical flow of foreground objects identified by TMO-generated [12] segmentation masks. Translational (Trans) and rotational (Rot) errors are computed by comparing estimated camera parameters using VGGSfM [100] with the ground truth. Geometric Consistency (Geo) is computed as the successful ratio of VGGSfM to estimate camera parameters. This indicates the quality of 3D geometry consistency of the generated scene. Appearance Consistency (Apr) is computed by comparing the cosine distance of each frame's CLIP [70] vision embedding to the average of the entire video clip.

## H   Societal Impacts

Our work proposes a new approach for real-time streaming video generation for interactive applications. Our approach is faster and more computationally efficient than existing approaches. This potentially enables the adoption of more real-time interactive applications. We do not consider our work to bring risk for significant negative societal impacts. The videos generated by our method still contain imperfections that are easy to identify as generated videos, which prevents the technology from being used for malicious purposes.

