# OpenReview forum: "Autoregressive Adversarial Post-Training for Real-Time Interactive Video Generation"
_NeurIPS.cc/2025/Conference — NeurIPS 2025 poster_

### Official Review · Reviewer_XbHv · 2025-07-02

**Clarity:** 3
**Significance:** 3
**Originality:** 2
**Rating:** 5
**Confidence:** 3

**Summary:**

The paper presents a way to turn a pre-trained video diffusion model, trained to generate multiple frames simultaneously to create a fixed length video, to an auto-regressive model that generates the next frame conditioned on the N previous frames.

The method consists in 3 steps:
* First the model is adapted to take as input not just the noise, but the concatenation of the previous frame to the noise (channel wise). The model is also adapted to use causal attention (frame by frame), turning it to an auto-regressive model.
* Then the model is distilled to generate the next frame in one step
* Then an adversarial loss is used to turn that frame into a higher quality frame, and student-forcing is used so that inference and training do not suffer from a mismatch in distribution that would make inference diverge quickly

Thanks to those adaptions, the authors claim that:
1) the model can generate 24fps real time, with a resolution of 736×416 on a single H100 GPU and 1280×720 on eight H100 GPUs
2) the model can generate minute long videos thanks to its auto regressive nature

**Questions:**

n/a

**Ethical Concerns:**

["NO or VERY MINOR ethics concerns only"]

**Quality:**

2

**Strengths And Weaknesses:**

Although the idea of one step distillation using GAN loss is not new, this paper introduces some nice tricks that make it better and combine it with auto-regressive generation frame by frame, to make video generation real time.

Strengths are:
- Great results in terms of generation speed
- An interesting combination of tricks to speed things up and push quality further (like frame by frame GAN loss, or student forcing)
- Convincing results, particularly for long 1 minute videos

Weaknesses:
- The evaluation protocol is relying on VBench-I2V which might not be the best correlated with human judgement on recent models, on qualitative evaluations, but no human evaluations (the golden standard)
- The benefits of the methods are most seen on long videos, but the videos shown are also very repetitive (there can be no scene change or interesting motion that change the layout too much with this approach)

---

> ### Author Rebuttal · Authors · 2025-07-30
>
> We sincerely appreciate the reviewer for the recognition of our work. The reviewer did not post any questions, but we acknowledge and agree with the weaknesses that the reviewer has pointed out. These are indeed issues that we aim to resolve and do better in future works. We thank the reviewer for the positive assessment of our work.

---

### Official Review · Reviewer_RpkT · 2025-07-02

**Clarity:** 3
**Significance:** 3
**Originality:** 2
**Rating:** 4
**Confidence:** 3

**Summary:**

This paper focuses on enabling real-time, interactive video generation by proposing Autoregressive Adversarial Post-Training (AAPT), a framework that transforms a pre-trained bidirectional latent video diffusion model into a fast, autoregressive generator. The method integrates block-causal transformer architecture with adversarial training, allowing the model to generate an entire latent frame in a single forward pass while using KV cache for efficiency. To mitigate error accumulation, the authors employ student-forcing adversarial training. Experiments show competitive quality and strong controllability in interactive tasks like pose-driven human and camera-controlled scene generation.

**Questions:**

Please refer to the Weaknesses above.

**Ethical Concerns:**

["NO or VERY MINOR ethics concerns only"]

**Final Justification:**

I appreciate the clarifications provided, and I will be keeping my score unchanged.

**Limitations:**

Yes

**Quality:**

3

**Strengths And Weaknesses:**

Strengths
1. The proposed method achieves real-time video generation at 24fps and high resolution (736×416 on 1×H100; 1280×720 on 8×H100) using only a single forward pass per latent frame, significantly outperforming existing methods in terms of latency and throughput.
2. By combining KV cache reuse and a segment-based training strategy, the model is capable of generating one-minute-long videos without stopping, which is notably longer than previous autoregressive diffusion models.
3. The model demonstrates competitive or state-of-the-art results across three tasks: standard I2V generation (VBench), pose-conditioned human generation, and camera-controlled world exploration, showing both high visual quality and controllability.

Weaknesses
1. The method extends Diffusion Adversarial Post-Training (APT) [1] to the autoregressive setting, but the novelty is incremental and not clearly disentangled. The key contributions (i.e., causal frame-level generation, student-forcing adversarial training, and efficient use of KV cache) are reasonable adaptations rather than fundamentally new ideas. Besides, the architecture builds heavily upon APT (as the authors themselves note that “Our method extends adversarial post-training (APT) to the autoregressive video generation scenario”). Thus, a more explicit comparison (both conceptual and empirical) to APT would help clarify what is truly novel.
2. The paper introduces a sliding window attention mechanism to limit memory and compute during long video generation, but it does not explore how the choice of window size $N$ impacts performance. A sensitivity analysis on $N$ would clarify the trade-offs between computational efficiency, temporal consistency, and generation quality.
3. While the paper reports strong performance on VBench-I2V and other quantitative metrics, it lacks human evaluation of perceptual quality and consistency, especially for the long video and interactive scenarios. Given that many metrics (e.g., FVD, aesthetic scores) are known to be imperfect proxies for human judgment, user studies would strengthen the paper’s claims on quality in real-time applications.

[1] Diffusion adversarial post-training for one-step video generation.

---

> ### Author Rebuttal · Authors · 2025-07-30
>
> We appreciate the reviewer for the thorough review. We address all the questions below.
>
> **1. Originality and novelty**
>
> In addition to the autoregressive adversarial training, our paper proposes several other novel contributions worth considering: architectural design (a recycle generator that fully exploits KV cache, and a parallel causal discriminator that produces per-frame logits), full student-forcing training (gradient pass through recurrent KV state), and long video training (through chunk-based generation and discrimination to address data and memory issues). Specifically, long video training enables us to be the first to achieve minute-long non-stop video generation fully using KV cache, and student-forcing shows significant improvement in quality degradation compared to diffusion-forcing methods.
>
> **2. Ablation study on window size**
>
> Our preliminary experiments find that increasing the sliding window size (from 1 to 5 seconds) significantly improves temporal consistency. This makes sense as the model can attend a farther distance. Therefore, for our final model, we choose to use the maximum window size that can fit the real-time and memory budget (5 seconds for 480p and 2.5 seconds for 720p).
> We have considered adding the ablation study to the window size. However, we realize that we do not have the resources to train multiple models of different window sizes. We have attempted to train one model where the window size is uniformly drawn between 1~5 seconds per sample during training. However, we find that the model trained this way has the best performance at inference when the window size is set to 2.5s, which is the median of the range. Therefore, this experiment is not convincing.
>
> We would like to emphasize that our paper focuses on the overall algorithmic contributions. The sliding window design is only a simple choice, and we leave exploration for better architecture for future study (discussed in limitations).
>
> **3. Human evaluation**
>
> Unfortunately, due to limited time and resources, we were not able to perform human studies before the deadline of NeurIPS submission. Therefore, we opted for more standardized evaluation metrics, VBench-I2V, because this allows us to compare with previous closed-source models (CausVid, which also reports VBench-I2V) and allows future works to compare with us.

---

> > ### Comment · Reviewer_RpkT · 2025-08-05
> >
> > I appreciate the clarifications provided, and I will be keeping my score unchanged.

---

### Official Review · Reviewer_oRaR · 2025-07-02

**Clarity:** 3
**Significance:** 3
**Originality:** 2
**Rating:** 4
**Confidence:** 5

**Summary:**

This paper introduces Autoregressive Adversarial Post-Training (AAPT), a method to transform pre-trained latent video diffusion models into efficient autoregressive generators for real-time interactive applications. The core innovation lies in leveraging adversarial training to enable single-step, per-frame generation (1NFE) while mitigating error accumulation through student-forcing training.

**Questions:**

**Questions about the "real-time"**

*1. The lightweight VAE decoder requires a more detailed explanation.*

Ablation studies by module are needed to clarify changes in the model's inference speed. Appendix E mentions that the lightweight VAE decoder achieves a 3x speedup compared to the original decoder. Therefore, it is necessary to demonstrate that the model's real-time performance stems from the main innovation(consistency distillation, adversarial training...) rather than other tricks.

*2. Does the rotary position embeddings (RoPE) impact inference speed?​​*

This work adopts the ​​MMDiT design​​ and employs a ​​sliding window​​ approach for long-video generation, making ​​RoPE particularly important​​ for maintaining positional awareness. The introduction of RoPE brings additional computational overhead. Please clarify how the inclusion of the ​​RoPE module​​ affects the model's inference speed.

*3. More quantitative/qualitative results are needed to demonstrate the effectiveness of post-training.*

This work has 3 stage (Diffusion adaptation, Consistency distillation, Adversarial training), but it only compares base diffusion models with 3-stage models, it does not clarify the performance changes at each stage. Many models claim to achieve real-time generation (SkyReel-2, CausVid, Matrix, GameNGen), with most employing distillation techniques (consistency distillation, DMD, etc.). However, in practice, distilled models (1-step, 4-step) often underperform the original model (50-step) in terms of human perceptual quality. This paper utilizes post-training to enhance visual fidelity and reduce error accumulation, but it lacks ablation studies to validate the performance of consistency distillation at 1-step, 4-step, and 8-step settings. Additionally, there is a lack of ablation experiments and visual comparisons (unless demonstrated in the project webpage/videos—please point them out if available) to substantiate the visual improvements brought by **Adversarial Post-Training**.

*4. Why autoregressive step is more efficient than onestep diffusion forcing？*

The paper mentions that the "recycling approach is more efficient than one-step diffusion forcing," yet diffusion forcing can also employ techniques like sliding windows (as in SkyReel-2) and KV caching (e.g., Magi-1, Playable Game Generation, and later works such as VideoMAR and self-forcing). Could the authors clarify the specific differences in how KV caching operates under these two guidance schemes?

**Questions about the "interactive"**


​​*5. Underperformance in Pose-Conditioned Human Video Generation.​​*

All metrics of AAPT fall behind OmniHuman-1 in this task. While this is understandable given AAPT's novel motivation, it remains necessary to clarify the ​​unique advantages​​ of this work. Moreover, this work follows the same approach(extract and encode the human pose) as OmniHuman-1, yet its performance is consistently inferior—which is quite strange.

*6. The memory capability of the interactive process model needs clarification.*

Examples on the demo page show that the model exhibits certain memory lapses (e.g., a man in the desert, where background characters suddenly disappear and reappear when the camera angle changes). The original text uses a sliding window approach for interactive generation, lacking a module to record historical information. However, in the field of interactive generation—such as game world models (The Matrix, GameGenX, Genie) and 3D world generation (Wonderland, WonderWorld)—memory plays a particularly crucial role in interactive systems. Please further supplement this in the "Related Works" section.

---

Overall, this paper addresses a very interesting and novel problem. However, most of the algorithms are derived from the baseline APT, and the additional autoregressive step proposed in this paper requires further clarification of its advantages compared to diffusion-forcing to address my concerns about the algorithmic novelty. I hope the authors can resolve my concerns.

**Ethical Concerns:**

["NO or VERY MINOR ethics concerns only"]

**Final Justification:**

my concerns have been addressed.

**Limitations:**

Yes.

**Paper Formatting Concerns:**

Mistake in preparing the manuscript: a paragraph is misplaced (line 224 to line 227).

**Quality:**

3

**Strengths And Weaknesses:**

**Strengths**:
- This work is written with fluent and clear prose. Rigorous 3-stage training (diffusion adaptation → consistency distillation → adversarial training) enables stable 1NFE generation.
- Comprehensive evaluation across 120-frame and 1440-frame generation using VBench-I2V, outperforming diffusion-forcing baselines (CausVid, SkyReel-V2) in throughput/latency .
- Ablation studies validate design choices (e.g., student-forcing > teacher-forcing, recycling input necessity).

**Weaknesses**:
- User study is needed to demonstrate effectiveness, although it is not mandatory.If the work already includes human evaluation, please specify details like the number of participants and evaluation methods.
- Mistake in preparing the manuscript: a paragraph is misplaced (line 224 to line 227), as the author has clarified in the appendix.
- More concerns about the "real-time" and "interactive", see the **Questions**.

---

> ### Author Rebuttal · Authors · 2025-07-30
>
> We appreciate the reviewer for the thorough review. We address all the questions below.
>
> **1. Lightweight VAE**
>
> For 736x416 resolution on an H100 GPU, at every autoregressive step, our DiT model takes about 0.120s to generate a latent frame, and our lightweight VAE takes about 0.041s to decode a latent frame (4 video frames).
>
> In other words, for a 60s 24fps video, the entire video can be generated in 58s (24.8 fps), where the one-step DiT cumulatively takes 43.2s (75%) and the lightweight VAE takes 14.7s (25%). Without the lightweight VAE, the full-size VAE would take 44s (3x), which would make it hard to fit the whole pipeline into a 60s budget. Similarly, without one-step DiT, the pipeline will not fit the budget either. Therefore, both the compressions on DiT and VAE parts are important, but DiT still takes most of the computation. We will add this detail to our final publication.
>
> **2. RoPE's impact on inference speed.**
>
> Our original bidirectional video diffusion model also uses RoPE following MMDiT. The use of RoPE is prevalent in recent video diffusion transformers (Hunyuan, WanX, Seaweed, Seadance, Mochi, etc.). Our work only modifies the frequencies used in RoPE (detailed in Appendix C). This modification is to support arbitrary-length training and inference. Our method's modification to RoPE does not incur additional cost on the inference speed relative to the base model.
>
> The use of RoPE does incur additional yet marginal per-layer computation compared to absolute positional encoding through input. However, this is similar to the QK norm, which can add inference cost but is commonly used for stable training. Our research focuses on the method of transforming an existing architecture into an autoregressive model. We leave the exploration of more efficient architecture or other possible kernel fusion optimizations to future work.
>
> **3. Effectiveness of adversarial post-training.**
>
> The reviewer asks for an ablation study between adversarial post-training and consistency distillation to show the effectiveness of adversarial post-training. We would like to clarify that this topic has been more thoroughly addressed in the APT work [1]. Specifically, the APT paper shows that consistency distillation alone generates very blurry results using only a single step, and adversarial post-training improves quality significantly (Figure 8 of the APT paper). The APT paper also shows quantitative VBench metrics comparing adversarial post-training and consistency distillation (Table 7 of the APT paper Appendix B). In our work, we observe a similar phenomenon where consistency distillation alone is insufficient for one-step generation (Our paper, Section 3.2). Since the effectiveness of APT is already established, we focus the ablation studies on the autoregressive generation (long-duration training, recycle input, student-forcing, etc.) which is our main contribution. We refer to the APT paper for more details.
>
> [1] Diffusion Adversarial Post-Training for One-Step Video Generation.
>
> **4. Why is our method more efficient than one-step diffusion forcing?**
>
> Figure 2 of our paper visualizes the difference between our method and one-step diffusion forcing.
> For example, for 4-step diffusion-forcing, we show the per-frame noise level below. The () indicates frames that must be recomputed on every autoregressive step. The frames outside of () can use KV cache.
> * [(1000)]
> * [(750, 1000)]
> * [(500, 750, 1000)]
> * [(250, 500, 750, 1000)]
> * [(0, 250, 500, 750, 1000)] - compute 5 frames
> * [0, (0, 250, 500, 750, 1000)] - compute 5 frames
> * [0, 0, (0, 250, 500, 750, 1000)] - compute 5 frames
>
> Notice how diffusion forcing needs to compute k+1 frames on every autoregressive step, where k is the number of denoising steps. Therefore, even if we push diffusion forcing to one step, it still must compute 2 frames on every autoregressive step.
> * [(1000)]
> * [(0, 1000)]
> * [0, (0, 1000)] - compute 2 frames
> * [0, 0, (0, 1000)] - compute 2 frames
>
> Fundamentally, this is because diffusion forcing recycles past prediction by token concatenation, whereas ours uses channel concatenation. When both using KV cache, our method only needs to compute 1 frame of tokens on every autoregressive step, whereas diffusion forcing needs to compute 2 frames of tokens.
>
> **5. Comparison to OmniHuman**
>
> We would like to clarify that neither OmniHuman-1 nor other comparison models in the pose-condition and camera-condition tasks are real-time interactive models. Specifically, OmniHuman-1 is a bidirectional video diffusion model that has similar parameters to ours but uses 25 sampling steps + CFG, so it is about 50 times slower than ours. We will improve the clarification in our final publication.
>
> **6. Long-memory architectures**
>
> Our paper mostly focuses on algorithmic contributions and adopts sliding windows for simplicity. We acknowledge the importance of exploring long-range memory architectures, and we leave them for future work (Section 5 discusses limitations). We appreciate the reviewer for pointing out relevant works. We will add them to our final publications.
>
> **7. Originality and novelty**
>
> Similar to the response to reviewer RpkT, we would like to point out here that, in addition to the autoregressive adversarial training, our paper proposes several other non-trivial contributions worth considering: architectural design (recycle generator that fully exploits KV cache, and parallel causal discriminator that produces per-frame logits), full student-forcing training (gradient pass through recurrent KV state), and long video training (through chunk-based generation and discrimination to address data and memory issues). Specifically, long video training enables our work to be, to the best of our knowledge, the first to achieve minute-long non-stop video generation fully using KV cache; student-forcing shows significant improvement in quality degradation compared to diffusion-forcing methods.
>
> We hope our response addresses all the concerns. Please let us know if there are any more questions, and we would be happy to address them. We hope the reviewer can reconsider improving the rating to accept our work.

---

> > ### Comment · Area_Chair_w8v3 · 2025-08-06
> >
> > I invite the reviewer to engage in the rebuttal discussion following author responses.

---

> > ### Comment · Reviewer_oRaR · 2025-08-06
> >
> > Thanks for the author's response; my concerns have been addressed. Considering the formatting errors in the manuscript and the unclear explanation about AR v.s. diffusion-forcing, I will raise my rating to 4.

---

### Official Review · Reviewer_RtHj · 2025-07-03

**Clarity:** 3
**Significance:** 3
**Originality:** 3
**Rating:** 5
**Confidence:** 4

**Summary:**

This paper tackles the challenge of real-time interactive video generation by introducing Autoregressive Adversarial Post-Training (AAPT), a method to convert a pretrained latent diffusion video model into a streaming one-step-per-frame generator. Instead of the usual multi-step diffusion sampling, AAPT produces each new video frame in a single network forward pass (1 NFE) by leveraging a transformer architecture with KV-cache for efficient autoregression. The model is trained with an adversarial objective after initial diffusion training, enabling high-fidelity frame synthesis and student-forced training (training on its own sequential outputs) to mitigate error accumulation over long sequences. Notably, it demonstrates non-stop video streaming up to 1 minute (1440 frames) without quality collapse, significantly outperforming prior one-step or diffusion-forcing methods that were limited to only a few seconds of footage.

**Questions:**

Training stability and compute cost: Adversarial post-training on an 8B parameter model is a substantial undertaking. Could the authors provide more insight into the training process? For instance, how long did AAPT training take, and were there any instability issues (mode collapse, divergence) that needed to be overcome? Understanding the computational cost and training stability would help gauge the practicality of reproducing or extending this work (e.g., applying AAPT to other base models or at larger scales).

**Ethical Concerns:**

["NO or VERY MINOR ethics concerns only"]

**Final Justification:**

I appreciate the authors' rebuttal and will keep my rating as Accept.

**Limitations:**

Yes

**Quality:**

3

**Strengths And Weaknesses:**

**Strengths:** The paper's technical quality is high, offering a novel yet well-founded solution to an important problem. The AAPT approach is **innovative** in combining diffusion models with **adversarial post-training** for video. It builds on recent **diffusion-forcing** ideas but extends them in a new direction. This design fully exploits the transformer's KV cache (like in LLMs) to avoid recomputation, yielding **significantly superior throughput and latency** suitable for real-time use. The empirical results are compelling: the model achieves **state-of-the-art speed** (24 fps at high resolution) far outperforming prior systems like CausVid (which reaches 9.4 fps at lower resolution). It also scales to much longer videos than most diffusion-based methods—*continuous 60+ seconds of video without resets*—whereas competitors like CausVid, SkyReel-V2, and MAGI-1 can only generate in 5-second chunks before needing to restart the generation process. AAPT improves per-frame image fidelity and condition consistency over the diffusion baseline, and achieves the **best overall quality scores** among the compared methods on long video generation. Overall, the authors demonstrate a strong grasp of both the engineering and research aspects, lending credibility to the **originality** of the solution (extending the recent APT idea into an autoregressive domain) and reinforcing the **importance** of the contribution.

**Weaknesses:** The weaknesses of the paper are relatively modest. One concern is that, as with many adversarial generation approaches, **temporal consistency** can suffer slightly; indeed, the authors report a minor drop in temporal quality metrics for AAPT compared to a pure diffusion baseline. There are instances of small **scene or object drift** over very long sequences (e.g., an object that disappears after occlusion, as noted by the authors), indicating that maintaining perfect identity and continuity remains challenging. The proposed solution (AAPT) still relies on a **sliding context window** for simplicity, meaning the generator only looks at a fixed recent history of frames; this could limit long-range coherence and is identified as an area for future improvement. Additionally, the training pipeline is quite complex—it involves diffusion fine-tuning, consistency distillation, and adversarial training with a large 8B parameter model. This raises the **computational cost** and implementation complexity: for example, adversarial post-training on such a large model likely required careful tuning (the paper doesn't detail training stability, which could be non-trivial). Finally, the experiments focus on the image-to-video setting (where the first frame is given); while this is a sensible choice for interactive applications, the method's performance in a pure text-to-video scenario (starting from scratch) is not explicitly demonstrated, a minor omission in scope.

---

> ### Author Rebuttal · Authors · 2025-07-30
>
> We sincerely appreciate the reviewer for the recognition of our work. We address the reviewer's questions below.
>
> **1. Training details**
>
> We have provided a more detailed training setup in "Appendix D - Computational Resources". Specifically, we used 256xH100 GPUs, and the entire AAPT I2V model is trained in approximately 7 days. The diffusion adaptation (stage 1) and long-video training (stage 3) consume the most time.
>
> We find that our training process is mostly stable after finding the correct hyperparameters (learning rate, batch size, r1 penalty scale, etc., as documented in the paper). Abeit, there are a few times the model diverges and we have to restart from the last proper checkpoints. Another thing to note is that for adversarial training, it is not longer training the better. We have to visualize the checkpoint to see if it reaches the peak quality before quality degrades. The training process still requires more babysitting. We are actively working on further improving the stability of the method.
>
> **2. Pipeline complexity**
>
> Another good point the reviewer points out is that the current pipeline still requires three stages. We also realize that this causes barriers in production, since every new task must start with diffusion training again. We are actively working on this.
>
> **3. Text-to-video generation**
>
> Our model architecture technically supports text-to-video generation by simply training it to accept an all-zero tensor as the first frame input. We have briefly experimented with it, and the model can indeed perform text-to-video generation. However, we did not choose to pursue this path because T2I one-step generation is still challenging. The APT paper [1] showed that one-step T2I/T2V generation still has a big quality gap. If our autoregressive model performs T2V, the first autoregressive step is equivalent to T2I one-step generation. We are actively working on this more fundamental problem, as improving one-step generation quality can improve our one-step streaming quality as well.
>
> [1] Diffusion Adversarial Post-Training for One-Step Video Generation.

---

### Official Review · Reviewer_szBj · 2025-07-05

**Clarity:** 4
**Significance:** 4
**Originality:** 4
**Rating:** 5
**Confidence:** 4

**Summary:**

The paper introduces a method that converts a pre-trained bidirectional video diffusion model into a causal generator of a single function evaluation per latent frame, enabling low-latency, real-time video generation.
The key contributions include model architecture and training procedure.
* In terms of model architecture, the bidirectional attention is replaced by block-causal attention with a sliding window. The clean previous frame is concatenated with noise as input, halving the computation compared to diffusion forcing.
* In terms of training procedure, it involves first finetuning the adapted architecture with teacher-forcing, then performs consistency distillation, and finally adversarial training. The majority of the innovations are concentrated in the adversarial training stage:
    * The discriminator for the adversarial training stage is initialized from the same generator weights, adapted to output a per-frame logit.
    * The student predictions are used as conditioning (student-forcing) instead of ground-truth frames (teacher-forcing), reducing error accumulation and encouraging self-improvement.
    * To generate long videos beyond the training data, the discriminator is applied in an overlapped, chunk-wise manner. To save memory when training for long video generation, the KV cache of the previous chunk is detached.

This approach achieves state-of-the-art performance at significantly higher speeds. Additionally, it can be fine-tuned to incorporate conditioning signals for human pose and camera control.

**Questions:**

* How is student-forcing implemented during adversarial training? It seems that the parallel prediction technique of LLM won’t work for teacher-forcing due to the dependency on previous predictions. Do you pre-sample the student outputs, or chain the sampling process and back-propagate through time?
* What will happen if the video is generated beyond 60 seconds? How does the failure mode look like?

**Ethical Concerns:**

["NO or VERY MINOR ethics concerns only"]

**Limitations:**

yes

**Quality:**

4

**Strengths And Weaknesses:**

### Strength
* The proposed method excels across multiple dimensions: high-resolution, high-quality, low-latency, and real-time. These are remarkable achievements.
* The paper is well written, with a comprehensive background introduction and has every important detail documented.
* Utilizing student-forcing to fundamentally reduce error accumulation is a principled idea that is well-implemented in the proposed model.
* Compared to the concurrent work self-forcing [1], the proposed method is simpler to implement and potentially faster, thanks to the 1-step generation design.
[1] Huang et al., Self Forcing: Bridging the Train-Test Gap in Autoregressive Video Diffusion, 2025

### Weaknesses
* While the proposed method demonstrates strong performance in terms of visual quality and speed, its reliance on sliding window for long-video generation is fundamentally constrained in modeling long-range dependencies. Consequently, it is not clear if the proposed approaches are viable as long-term solutions.
* There are no visual results provided in the supplementary material.

---

> ### Author Rebuttal · Authors · 2025-07-30
>
> We sincerely appreciate the reviewer for the recognition of our work. We address the reviewer's questions below.
>
> **1. Visual results**
>
> We have actually provided visual results in supplementary material through an external website. Specifically, Appendix B provided a link to an anonymized website that has all the visual samples and comparisons. We have also provided a five-minute (7200 frame) result in the last section of the page ("Toward Infinite-Length Streaming"). This shows our model's behavior when generating beyond 60 seconds.
>
> **2. Student-forcing implementation**
>
> During student-forcing adversarial training, the generator generates samples in the exact same way as inference. Specifically, given the first frame (I2V first frame is taken from the dataset), the model recurrently generates the next frame. The frame is recycled as input for the next autoregressive step, along with the KV cache. This recurrent for-loop stops when the target length is reached. This produces the whole video clip.
>
> During training, the past frame prediction output is detached before recycling as input for the next autoregressive step. However, the KV state is not detached. So during the backward pass, the gradient can still flow backward through time (through the KV state) to reach all parameters.
>
> **3. The use of a sliding window**
>
> Our work focuses on algorithmic innovation (autoregressive adversarial training, student-forcing, long-video training). We adopted the sliding window technique because it incurs minimal change to the base model. We leave it to future papers to explore long-range memory architectures (for example, inserting a recurrent memory layer, etc.)

---

> > ### Comment · Reviewer_szBj · 2025-08-08
> >
> > I would like to thank the authors for the clarification. I checked the videos -- they look great!

---

### Comment · Area_Chair_w8v3 · 2025-08-04
**Discussion engagement**

Hello all. Thanks to all reviewers and authors for their implication in the review process so far. If not already done, we encourage all reviewers to engage with authors following rebuttals since the discussion period is coming to a close in a few days. Thank you for helping ensure a quality  NeurIPS review process!

---

### Decision · Program_Chairs · 2025-09-17

**Decision:**

Accept (poster)

**Comment:**

The paper explores a method to leverage pre-trained video diffusion models for real-time video generation. The reviewers found the method to be novel, sound, and an improvement on existing approaches.